# Effects of Pelleting and Long-Term High-Temperature Stabilization on Vitamin Retention in Swine Feed

**DOI:** 10.3390/ani12091058

**Published:** 2022-04-20

**Authors:** Huakai Wang, Longxian Li, Nan Zhang, Tuan Zhang, Yongxi Ma

**Affiliations:** State Key Laboratory of Animal Nutrition, College of Animal Science and Technology, China Agricultural University, Beijing 100193, China; huakaiwhk@cau.edu.cn (H.W.); s20193040586@cau.edu.cn (L.L.); s20203040629@cau.edu.cn (N.Z.); tuan0900@163.com (T.Z.)

**Keywords:** vitamin, pelleting, long-term high-temperature stabilization, retention

## Abstract

**Simple Summary:**

Pelleting of animal feeds has been practiced for decades, to denature protein, increase starch gelatinization, decrease anti-nutrient factor content, and improve the nutritional value of feed ingredients. Long-term high-temperature stabilization after pelleting has been employed by feed manufacturers to control the spread of pathogenic microorganisms, especially African swine fever through feed. However, pelleting is an aggressive process against vitamins, and prolonged high temperatures may further damage the vitamins’ structure. Our results showed that the recovery of vitamin A, vitamin E, vitamin B_2_, and vitamin B_6_ significantly decreased after pelleting and long-term high-temperature stabilization, and the high-temperature stabilization process has the most significant influence.

**Abstract:**

The objective of this study was to study the effect of pelleting and long-term high-temperature stabilization on the retention of vitamin A, vitamin E, vitamin B_2_, and vitamin B_6_ in swine feed. Piglet diets (diet 1 and 3) were pelleted after conditioning at 83 °C for 120 s, and were high-temperature stabilized at 90 °C for 8.5 min after pelleting; the finishing pig diets (diet 2, 4, and 5) were pelleted after conditioning at 82 °C for 90 s, and were high-temperature stabilized at 85 °C for 9 min after pelleting; the samples were obtained before condition, after condition, after pelleting, and after cooling. The contents of vitamin A and vitamin E in diets 1–5 and vitamin B_2_, and vitamin B_6_ in diets 3–5 were detected. The results showed that: (1) the conditioning process had no significant effect on the retention of vitamin A, vitamin E, vitamin B_2_, and vitamin B_6_ in all experimental diets (*p* > 0.05); (2) the pelleting process and high-temperature stabilization process after pelleting had different degrees of influence on vitamins, among which the stabilization process had a more significant effect on the retention of vitamins. After pelleting and long-term high-temperature stabilization, the retention of vitamin A, vitamin E, and B_2_, and vitamin B_6_ were 68.8–77.3%, 56.9–90.1%, 63.8–70.3%, and 60.1–67.0%, respectively. In the process of pelleting and long-term high-temperature stabilization, the retention of vitamin A, vitamin E, vitamin B_2_, and vitamin B_6_ in the feed were significantly reduced (*p* < 0.05). Therefore, vitamin loss during high temperature and over a long period of time is worth considering, and vitamins must be over-supplemented.

## 1. Introduction

African Swine Fever (ASF) is an acute, febrile, and high-contact infectious disease caused by a highly lethal and infectious virus (African Swine Fever Virus, ASFV), which has caused great harm to the pig industry [1,2]. Facing the challenge of ASFV, major breeding enterprises actively take various measures to enhance feed safety and reduce the risk of ASFV infection. The ASFV has been reported to be inactivated by heating at 70 °C for 30 min or at 85 °C for 5 min [3]. Therefore, “ultra-long high-temperature stabilization after pelleting process” may be a way to reduce ASFV infection through feed, mainly by subjecting the feed to a curing process of 80–110 °C after pelleting. However, temperature, humidity, pressure and friction in the pelleting process will affect the activity of nutrients in the feed, especially those with high sensitivity such as vitamins [4,5]. Vitamins are a small number of organic compounds needed to maintain life and maintain normal physiological function, growth and development, life function, health and reproduction, etc., and are essential for metabolism [6]. High-temperature pelleting should not only increase the safety of feed, but also supplement sufficient amount of trace substances such as vitamins to prevent animals from being in a sub-health state for a long time. However, there is a shortage of information on vitamin loss after pelleting and long-term high-temperature stabilization regarding feed processing used in the swine industry to prevent the spread of African swine fever. Therefore, the aim of this study was to investigate the vitamin loss caused by the long-term high-temperature treatment of feed by detecting the content of vitamin A, vitamin E, vitamin B_2_, and vitamin B_6_ before condition, after condition, after pelleting, and after high-temperature stabilization.

## 2. Materials and Methods

The experimental diets were prepared and pelleted in the China Agricultural University Feed Mill Educational Unit (Beijing, China). The samples were tested in the Ministry of Agriculture and Rural Affairs Feed Efficacy and Safety Evaluation Center, China Agricultural University (Beijing, China).

### 2.1. Experimental Design and Diets

One-way variance design was used in this experiment. Five diets were used in this study: diet 1 (piglet feed, basal), diet 2 (growing pig feed, basal), and diet 3 (piglet feed, based on diet 1, supplemented with vitamin A, E, B_2_, and B_6_ at the level of 22,000 IU/kg, 300 mg/kg, 150 mg/kg and 180 mg/kg, respectively), diet 4 (growing pig feed, based on diet 2, an additional 12,000 IU/kg vitamin A, 350 mg/kg vitamin E, 150 mg/kg vitamin B_2_ and 180 mg/kg vitamin B_6_ were added), and diet 5 (growing pig feed, supplemented with 32,500 IU/kg vitamin A, 100 mg/kg vitamin E, 20 mg/kg vitamin B_2_, and 70 mg/kg vitamin B_6_ based on diet 4). The adding levels and measured levels of vitamin A, E, B_2_, and B_6_ are shown in Table 1. The ingredients and chemical composition of experimental diets are shown in Table 2.

### 2.2. Processing Parameters

Three batches (100 kg each batch) of each diet were mixed using a horizontal single-ribbon mixer (Model SLHY 2.5c, Muyang, Yangzhou, China) for 300 s, and pelleted using a pellet mill (Famsun 180, Yangzhou, China), and each batch represented one repetition. The pelleting parameters of experimental diets are shown in Table 3. The samples were obtained from four locations, including before conditioning, after conditioning, after pelleting, and after cooling, and seven repeated samples (100 g each sample) were taken from each position in chronological order. All samples were crushed to 1 mm using a high-speed universal crusher (FW-100, Beijing, China) for further laboratory analysis.

### 2.3. Chemical Composition Analysis

Vitamins A (VA) and E (VE) were detected by high-performance liquid chromatography (HPLC) according to the AOAC method [7]. In brief, a 2 g sample was dissolved in the enzyme solution at 37 ± 2 °C water bath and extracted with methanol. The extract was filtered through a 0.45 µm filter membrane for the HLPC system (Agilent 1200 Series; Agilent Technologies Inc., Santa Clara, CA, USA). Vitamin B_2_ and Vitamin B_6_ were extracted from diets according to the procedure of Chen et al. with some modification [8]. A 5 g sample was extracted with phosphate buffer (PBS) and sonicated in a boiling water bath. The supernatants of the extracted samples were analyzed by liquid chromatography using a 250 mm × 4.5 mm × 5 μm, Eclipse Plus C18 column (Agilent Technologies Inc., Santa Clara, CA, USA) [9]. All samples were assayed in triplicate.

### 2.4. Calculation of Vitamin Retention Rate

Vitamin retention rate (%) = (vitamin content at different stages sample/vitamin content before conditioning) × 100 [10].

### 2.5. Statistical Analysis

The normality of data (vitamin retention each sampling point) was verified using the UNIVARIATE procedure of SAS (SAS Institute, Cary, NC, USA), and the BOXPLOT procedure of SAS was used to check for outliers. One-way ANOVA was conducted by GLM process in SAS 9.4 statistical software, and multiple comparisons were conducted by Turkey method. *p* < 0.05 was considered a significant difference.

## 3. Results

### 3.1. Retention of Vitamin A

As shown in Table 4, vitamin A content in all experimental diets continued to decrease during conditioning, pelleting, and stabilization after pelleting, and the retention of vitamin A after cooling was 68.8–77.3%. The vitamin A retention in diets was not significantly affected by conditioning (*p* > 0.05) but was significantly affected by pelleting and stabilization process after pelleting (*p* < 0.05). The vitamin A retention in diets 1 and 2 was significantly higher than in diets 3–5 after pelleting and cooling (*p* < 0.05).

### 3.2. Retention of Vitamin E

As shown in Table 5, the vitamin E retention continued to decrease during the pelleting process except the diet 2, and the final retention was 56.9–85.9%. There had no significant effect on the vitamin E retention in diets after conditioning (*p* > 0.05). However, the vitamin E retention in diets 3 and 5 was significantly decreased during the pelleting process (*p* < 0.05), and the vitamin E retention in diets 3, 4, and 5 was significantly decreased during the stabilization process after pelleting (*p* < 0.05). In addition, the vitamin E retention in diet 5 was significantly lower than that in diets 1–4 after cooling (*p* < 0.05).

### 3.3. Retention of Vitamin B_2_

As shown in Table 6, the retention of vitamin B_2_ in all experimental diets was significantly decreased during high-temperature pelleting (*p* < 0.05), and the retention was 63.8–70.3% after cooling. Conditioning had no significant effect on vitamin B_2_ retention in all diets (*p* > 0.05), but pelleting decreased the vitamin B_2_ retention in diets 5 (*p* < 0.05), and stabilization process after pelleting decreased the vitamin B_2_ retention in diets 3–5 (*p* < 0.05). In addition, the retention of vitamin B_2_ in diet 5 was lower than that in diets 3 and 4 after pelleting and significantly higher than that in diets 3 and 4 after cooling (*p* < 0.05).

### 3.4. Retention of Vitamin B_6_

As shown in Table 7, the retention of vitamin B_6_ in all experimental diets was 60.1–67.0% after cooling. There was no significant effect on vitamin B_6_ retention during conditioning and pelleting process (*p* > 0.05), but the stabilization process after pelleting significantly reduced vitamin B_6_ retention (*p* < 0.05), and the vitamin B_6_ retention in diet 3 after cooling was significantly higher than that in diets 4 and 5 (*p* < 0.05).

## 4. Discussion

The traditional feed-pelleting process mainly includes the conditioning and pelleting process. To cope with the spread of ASF, most feed processing enterprises add a long-term high-temperature stabilization process after pelleting based on the traditional pelleting process to achieve the double insurance of killing ASFV [11]. The supplemental level of vitamin in feed is usually 10–20% higher than the actual requirement to compensate for vitamin loss during the traditional pelleting process. In contrast, the long-term high-temperature stabilization process causes more significant vitamin loss, so it might still be insufficient to meet the needs of animals. This experiment tested the retention of vitamins A, E, B_2_, and B_6_ in pelleting and the long-term high-temperature stabilization process (85–90 °C, 8.5–9 min) to guide feed-production enterprises to provide sufficient vitamins for animals.

Conditioning mainly involves humidifying and heat treatment to soften feed materials, improving their chemical properties, reducing energy consumption, and improving pelleting efficiency and particle quality [12]. This study showed that conditioning did not affect vitamin A, E, B_2_, and B_6_ retention, and their retentions were 93.0–98.3%, 91.7–98.5%, 90.9–98.6%, and 96.3–98.4% after conditioning. This result is partly consistent with the reports of Lewis et al. [13], who found that the retention of vitamin B_2_, niacin, and vitamin D_3_ was not affected by the conditioning time, the conditioning temperature, and their interaction. However, the conditioner’s high temperature and high humidity environment provides energy and medium for the redox reaction of vitamins and might promote their decomposition [14]. It has been interpreted that the conditioning processing does not affect vitamin stability as the conditioning time associated with animal feed pelleting does not appear to be long enough to affect vitamin concentration [13].

The stability of vitamins was affected by various factors, such as temperature, oxidation, abrasion, and moisture during pelleting [5]. The feed after conditioning is pressed into the die by the press roller and deformed under intense pressure, which might lead to the destruction of the structure of vitamins [15]. The rapid increase in temperature and pressure caused by pelleting also promotes the redox reaction of vitamins and decreases their activities [16]. In this experiment, the retention of vitamin A, E, B_2_, and B_6_ after pelleting were 74.6–89.1%, 65.9–92.3%, 87.9–96.9%, and 89.2–96.7%. The retention of vitamin E in diet 5 after pelleting (65.9%) was significantly lower than that in other diets, which may be related to the fact that the highest content of vitamin E in diet 5 is more easily destroyed under a high-temperature environment. The chemical structure of vitamin A contains conjugated double bonds and hydroxyl groups, which are more prone to oxidation, dehydration and decarboxylation under high temperature and high humidity conditions, resulting in worse stability and more straightforward reduction of its biological activity [17]. Marchetti et al. [14] pointed out that the loss rate of vitamin A reached nearly 50% after pelleting; at the same time, the loss rate of vitamin B_2_ and B_6_ was much lower than that of vitamin A, which is in agreement with our previous study [18].

The traditional pelleting process has the greatest loss of vitamin in the process of conditioning and granulation. We simulate the post-curing process used by feed manufacturers in response to ASF by adding a long-term high-temperature stabilization process after pelleting in this experiment. The results showed that the retention of vitamin A, E, B_2_, and B_6_ decreased significantly after the high-temperature stabilization process. In addition, the loss of vitamin A and E in the stabilization process did not exceed the total loss rates of conditioning and pelleting, while the loss of vitamin B_2_ and B_6_ in the stabilization process exceeded the total loss rates of conditioning pelleting. In the process of high-temperature stabilization, the long-term high-temperature environment may promote the occurrence of the Maillard reaction, hydrolysis reaction, lipid, Fenton type or microbial-induced redox reaction of vitamins [19,20], resulting in more serious loss of vitamins in pellet feed during conditioning and pelleting in the process of high-temperature stabilization. Therefore, according to the results of the study, if, in accordance with the previous way of adding 10–20% vitamins in the feed, after ultra-long high-temperature stabilization, the effective activity of most vitamins is far lower than the animal requirements.

## 5. Conclusions

The retention of vitamin A, vitamin E, vitamin B_2_, and vitamin B_6_ decreased significantly after pelleting and long-term high-temperature stabilization, and the high-temperature stabilization process has the most significant influence. After pelleting and long-term high-temperature stabilization, the retention of vitamins A, E, B_2_, and B_6_ were 68.8–77.3%, 56.87–90.07%, 63.8–70.3%, and 60.1–67.0%, respectively. We suggest that the supplemental level of vitamins should refer to the retention of the vitamins in feed after different processing to meet the needs of animals for vitamin nutrition.

## Figures and Tables

**Table 1 animals-12-01058-t001:** Vitamins content in the diets before condition (air-dry basis).

Diets	VA (IU/kg)		VE (mg/kg)		VB_2_ (mg/kg)		VB_6_ (mg/kg)	
	Adding Level	Measured Level	Adding Level	Measured Level	Adding Level	Measured Level	Adding Level	Measured Level
1	8250.0	8779.8	80.0	83.6	5.0	-	25.0	-
2	8500.0	8690.3	85.0	94.3	10.0	-	25.0	-
3	30,250.0	31,375.6	380.0	403.3	200.0	161.5	205.0	212.3
4	20,500.0	18,238.2	435.0	435.6	160.0	167.0	205.0	196.9
5	53,000.0	55,352.8	535.0	529.6	180.0	185.6	275.0	265.5

**Table 2 animals-12-01058-t002:** Ingredients and chemical composition of experimental diets (air-dry basis).

Ingredients, %	Piglet Diet, Basal	Growing Pig Diet, Basal
Corn	29.70	65.70
wheat	10.00	-
Puffed corn	20.00	-
Soybean meal (46%)	17.30	16.00
DDGS ^1^	-	6.00
Wheat bran	-	8.00
Fish meal	3.30	-
Whey powder	7.50	-
Fermented soybean meal	5.00	-
Limestone	0.70	1.00
Dicalcium phosphate	0.80	0.60
L-Lysine-HCl (98%)	0.50	0.50
L-Threonine	0.20	0.20
Premix ^2^	5.00	-
Premix ^3^	-	2.00
Total	10.00	-
Nutrient level ^4^, %	100.00	100.00
Digestible energy, kcal/kg	3280.00	3250.00
Dry matter	89.00	88.00
Crude protein	19.00	16.00
Ether extract	2.02	2.95
Crude fiber	2.46	2.90
Ash	5.23	4.47
Lysine	1.30	0.95
Methionine + cysteine	0.81	0.62
Threonine	0.85	0.70
Calcium	0.65	0.6
Total phosphorus	0.55	0.46

^1^ DDGS: Distillers Dried Grains with Solubles. ^2^ The premix provided the following (per kilogram of compound feed): vitamin A, 8250 IU; vitamin D_3_, 825 IU; vitamin E, 80 IU; niacin, 35 mg; D-pantothenic acid, 15 mg; vitamin B_2_, 5 mg; vitamin K, 4 mg; vitamin B_6_ 25 mg; folic acid, 2 mg; thiamine, 1 mg; biotin, 0.2 mg; and vitamin B_12_, 0.025 mg. Zn, 125 mg; Fe, 75 mg; Cu, 50 mg; Mn, 25 mg; I, 0.5 mg; and Se, 0.1 mg. ^3^ The premix provided the following (per kilogram of compound feed): vitamin A, 8500 IU; vitamin D_3_, 1100 IU; vitamin E, 85 IU; niacin, 35 mg; D-pantothenic acid, 15 mg; Vitamin B_2_, 10 mg; menadione, 4 mg; Vitamin B_6_ 25 mg; folic acid, 2 mg; thiamine, 1 mg; biotin, 0.2 mg; and vitamin B_12_, 0.025 mg. Zn, 100 mg; Fe, 100 mg; Cu, 15 mg; Mn, 20 mg; I, 0.21 mg; and Se, 0.15 mg. ^4^ Nutrient levels are calculated.

**Table 3 animals-12-01058-t003:** Pelleting parameters of experimental diets.

Diets	Machine Type	RA, mm	LD	CT, °C	PT, °C	ST, °C	CHT, s	ST, Min
1	MUZL180	3.0	4:1	83.1	88.5	90	120	8.5
2	MUZL180	4.0	9:1	82.4	91	85	90	9.0
3	MUZL180	3.0	4:1	83.2	88.4	90	120	8.5
4	MUZL180	4.0	9:1	82.1	90.5	85	90	9.0
5	MUZL180	4.0	9:1	82.3	90.5	85	90	9.0

RA: Ring mode aperture, mm; LD: Length to diameter ratio; CT: Condition temperature, °C; PT: Pelleting temperature, °C; ST: Stabilizer temperature, °C; CHT: Conditioning holding time, s; ST: Stabilization time, min; MGZ: Mean grain size, mm.

**Table 4 animals-12-01058-t004:** The retention of vitamin A in diets during pelleting and the long-term high-temperature stabilization process.

Diets	Vitamin A Retention/%				SEM	*p*-Value
	Before Condition	After Condition	After Pelleting	After Cooling		
1	100.0 ^A^	93.0 ^A^	85.2 ^B,a^	74.9 ^C,a^	0.41	<0.05
2	100.0 ^A^	93.9 ^A^	89.1 ^A,a^	77.3 ^B,a^	0.46	<0.05
3	100.0 ^A^	98.3 ^A^	76.3 ^B,b^	68.8 ^C,b^	0.61	<0.05
4	100.0 ^A^	96.2 ^A^	74.6 ^B,b^	69.5 ^C,b^	0.58	<0.05
5	100.0 ^A^	95.4 ^A^	75.5 ^B,b^	69.2 ^B,b^	0.72	<0.05
SEM		0.15	0.22	0.14		
*p*-value		0.26	<0.05	<0.05		

^A,B,C^ Values with different capital letter superscripts in the same row mean significant difference (*p* < 0.05), ^a,b^ values with different small letter superscripts in the same column mean significant difference (*p* < 0.05).

**Table 5 animals-12-01058-t005:** The retention of vitamin E in diets during pelleting and the long-term high-temperature stabilization process.

Diets	Vitamin E Retention/%				SEM	*p*-Value
	Before Condition	After Condition	After Pelleting	After Cooling		
1	100.0 ^A^	91.7 ^A,B^	87.6 ^A,B,a,b^	85.9 ^B,b^	0.27	<0.05
2	100.0	94.3	92.2 ^a^	90.1 ^a^	0.27	0.34
3	100.0 ^A^	98.5 ^A^	85.3 ^B,b^	72.2 ^C,d^	0.53	<0.05
4	100.0 ^A^	95.8 ^A^	92.3 ^A,a^	80.1 ^B,c^	0.38	<0.05
5	100.0 ^A^	95.5 ^A^	65.9 ^B,c^	56.9 ^C,e^	1.02	<0.05
SEM		0.21	0.29	0.35		
*p*-value		0.38	<0.05	<0.05		

^A,B,C^ Values with different capital letter superscripts in the same row mean significant difference (*p* < 0.05), ^a,b,c,d,e^ values with different small letter superscripts in the same column mean significant difference (*p* < 0.05).

**Table 6 animals-12-01058-t006:** The retention of vitamin B_2_ in diets during pelleting and the long-term high-temperature stabilization process.

Diets	Vitamin B_2_ Retention/%				SEM	*p*-Value
	Before Condition	After Condition	After Pelleting	After Cooling		
3	100.0 ^A^	98.6 ^A,a^	96.9 ^A,a^	63.8 ^B,b^	0.75	<0.05
4	100.0 ^A^	91.5 ^A,b^	91.1 ^A,b^	64.3 ^B,b^	0.62	<0.05
5	100.0 ^A^	90.9 ^A,B,b^	87.9 ^B,b^	70.3 ^C,a^	0.47	<0.05
SEM		0.26	0.22	0.17		
*p*-value		<0.05	<0.05	<0.05		

^A,B,C^ Values with different capital letter superscripts in the same row mean significant difference (*p* < 0.05), ^a,b^ values with different small letter superscripts in the same column mean significant difference (*p* < 0.05).

**Table 7 animals-12-01058-t007:** The retention of vitamin B_6_ in diets during pelleting and the long-term high-temperature stabilization process.

Diets	Vitamin B_6_ Retention/%				SEM	*p*-Value
	Before Condition	After Condition	After Pelleting	After Cooling		
3	100.0 ^A^	98.3 ^A^	95.7 ^A,a^	67.0 ^B,a^	0.67	<0.05
4	100.0 ^A^	96.3 ^A^	89.2 ^A,b^	60.1 ^B,b^	0.81	<0.05
5	100.0 ^A^	98.4 ^A^	90.0 ^A,a,b^	62.0 ^B,b^	0.77	<0.05
SEM		0.19	0.25	0.23		
*p*-value		0.55	<0.05	<0.05		

^A,B^ Values with different capital letter superscripts in the same row mean significant difference (*p* < 0.05), ^a,b^ values with different small letter superscripts in the same column mean significant difference (*p* < 0.05).

## Data Availability

Not applicable.

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
