# Peer review of "Effects of Pelleting and Long-Term High-Temperature Stabilization on Vitamin Retention in Swine Feed"

_animals, 2022, doi:10.3390/ani12091058_

Round 1

Reviewer 1 Report

The paper entitled “Effects of pelleting and long-term high temperature stabilizing on vitamin retention in swine feed” was conducted to study the effect of pelleting and long-term high temperature stabilizing on the retention of vitamin A, vitamin E, vitamin B2, and vitamin B6 in swine feed. I do not recommend publication of the paper due to the following reasons:

  1. The introduction is not cohesive and lacks clarity. The objective of the study is not adequately justified. The objective of the study is not specific.
  2. The materials and methods lack clarity to the extend that the experiment cannot be repeated if the need arise. The experimental design is not known. The experimental conditions or feed processing parameter are not known even though the authors provided a title for processing parameter. For instance, what is the temperature and pressure for pelleting the experimental diets? What about conditioning? Etc. The control of the experiment was not stated. The treatments were not described. What was the sample size for each treatment? How many replications per treatment? How was the feed sample? What volume of feed was sampled?
  3. Authors indicated that vitamins were analyzed in lines 102 to 112. However, in the supplemental sheet (table S3), calculated values were shown. Why the contradiction?
  4. The statistical analysis was poorly described. What are the response variables? Was the data normally distributed? Did you remove any outlier from the data set?
  5. Some of the results presented are not true base on the table of results. For instance, from line 172 to 172, “vitamin B6 …………………………..high temperature pelleting” is not true base on table 5. Also, lines 175 to 176, “but the ……………………………………………..retention rate” is not true base on table 5. Additionally, lines 143 to 144, “and the vitamin E…………………………………..pelleting process” is not true for all diets.
  6. Missing superscript in table 3, for diet 2.
  7. The statement in line 120 to 121, “Draw………………………………….recovery rate ” is not clear.

Author Response

Dear Reviewer1:

Thank you very much for your critical comments on our manuscript. We have revised our manuscript according to your comments. The specific responses are listed below.

The paper entitled “Effects of pelleting and long-term high temperature stabilizing on vitamin retention in swine feed” was conducted to study the effect of pelleting and long-term high temperature stabilizing on the retention of vitamin A, vitamin E, vitamin B2, and vitamin B6 in swine feed. I do not recommend publication of the paper due to the following reasons:

Point 1: The introduction is not cohesive and lacks clarity. The objective of the study is not adequately justified. The objective of the study is not specific.

Response: Thanks for your comments. We have revised this portion.

Point 2: The materials and methods lack clarity to the extend that the experiment cannot be repeated if the need arise. The experimental design is not known. The experimental conditions or feed processing parameter are not known even though the authors provided a title for processing parameter. For instance, what is the temperature and pressure for pelleting the experimental diets? What about conditioning? Etc. The control of the experiment was not stated. The treatments were not described. What was the sample size for each treatment? How many replications per treatment? How was the feed sample? What volume of feed was sampled?

Response: Thanks for your comments. We have revised this portion.

Point 3: Authors indicated that vitamins were analyzed in lines 102 to 112. However, in the supplemental sheet (table S3), calculated values were shown. Why the contradiction?

Response: Thanks for your comments. We have revised this portion.

Point 4: The statistical analysis was poorly described. What are the response variables? Was the data normally distributed? Did you remove any outlier from the data set?

Response: Thanks for your comments. We have revised this portion.

Point 5: Some of the results presented are not true base on the table of results. For instance, from line 172 to 172, “vitamin B6 …………………………..high temperature pelleting” is not true base on table 5. Also, lines 175 to 176, “but the ……………………………………………..retention rate” is not true base on table 5. Additionally, lines 143 to 144, “and the vitamin E…………………………………..pelleting process” is not true for all diets.

Response: Thanks for your comments. We have revised this portion.

Point 6: Missing superscript in table 3, for diet 2.

Response: Thanks for your comments. We have revised this portion.

Point 7: The statement in line 120 to 121, “Draw………………………………….recovery rate ” is not clear.

Response: Thanks for your comments. We have revised this portion.

Reviewer 2 Report

  1. It is generally known that a pelleting condition (high temperature) negatively affect the retention of vitamin in feed. To evaluate the effect of long-term high temperature stabilizing, the traditional temperature condition (control) group is required to compare. Unfortunately, the control group is not appeared in this article.
  2. Discussion must be intensively done, particular the significant different between after pelting and after cooling.

Author Response

Dear Reviewer2:

Thank you very much for your effort for our work to improve our manuscript. The specific responses are listed below.

Point 1: It is generally known that a pelleting condition (high temperature) negatively affect the retention of vitamin in feed. To evaluate the effect of long-term high temperature stabilizing, the traditional temperature condition (control) group is required to compare. Unfortunately, the control group is not appeared in this article.

Response 1: Thanks for your comments. We aimed to study the effect of pelleting and long-term high temperature stabilization on the retention of vitamin A, vitamin E, vitamin B2, and vitamin B6 in swine feed, and the pelleting parameter is consistent with the traditional temperature condition, we only added an long-term high temperature stabilization after pelleting. In addition, we obtained the samples from different point, includes before conditioning, after conditioning, after pelleting, and after cooling. Therefore, we didn't have a separate control group.

Point 2: Discussion must be intensively done, particular the significant different between after pelting and after cooling.

Response 2: Thanks for your comments. We have revised this as suggested.

Reviewer 3 Report

The manuscript focus a topic of interest for the animal nutrition, with interesting results. However, there are some details that requires further description.

Please, provide changes and answers to the follwing minor comments:

Line 42,. Give data on needed temperatures

Line 43. & Line 187- describe the basis of the pelleting and long-term high temperature stabilizing process, and average conditions usually used.

Line 44-52  it is not relevant for the purpose of the study

Introduction lacks of an hypothesis that justify your objectives and also your experimental design, including the evaluation of increasing doses of vitamins in the feed and the measurement of vitamins at different step of the technological process.

Is it “retention” or “recovery”?

Line 93. Processing parameters are not described in this paragraph. Even that parameters are provided in Table S2, I would suggest giving main values in the text.

Line 195.- give average values of time and temperature

From line 207, provide discussion about the significant effects observed among diets related with the super doses of vitamins. Provide discussion about the sensitivity of different vitamins and the likely uses of microencapsulation forms to prevent these losses. You have previous articles on that.

Line 223-224. Describe the stabilizing process as compared to an standard pelletization

Line 225-228. This comparison among the response of different vitamins to the pelletization process and the stabilizing process merits a more detailed discussion and published references on the sensitivity of vitamins to heat and pressure treatments.

Tables: provide only 1 decimal for the average means, and 2 decimals for the SEM values (*alway one decimal more for the variability values than mean values)

Author Response

Dear Reviewer3:

Thank you very much for your effort for our work to improve our manuscript. The specific responses are listed below.

Point 1: Line 42,. Give data on needed temperatures

Response 1: Thanks for your suggestion. We have revised this as suggested.

Point 2: Line 43. & Line 187- describe the basis of the pelleting and long-term high temperature stabilizing process, and average conditions usually used.

Response 2: Thanks for your comments. We have revised this as suggested.

Point 3: Line 44-52 it is not relevant for the purpose of the study

Response 3: Thanks for your suggestion. We have revised this portion.

Point 4: Introduction lacks of an hypothesis that justify your objectives and also your experimental design, including the evaluation of increasing doses of vitamins in the feed and the measurement of vitamins at different step of the technological process.

Response 4: Thanks for your comments. We have revised this portion.

Point 5: Is it “retention” or “recovery”?

Response 5: Thanks. It is “retention”.

Point 6: Line 93. Processing parameters are not described in this paragraph. Even that parameters are provided in Table S2, I would suggest giving main values in the text.

Response 6: Thanks for your suggestion. We have revised this portion.

Point 7: Line 195.- give average values of time and temperature

Response 7: Thanks for your suggestion. We have revised this as suggested.

Point 8: From line 207, provide discussion about the significant effects observed among diets related with the super doses of vitamins. Provide discussion about the sensitivity of different vitamins and the likely uses of microencapsulation forms to prevent these losses. You have previous articles on that.

Response 8: Thanks for your suggestion. We have revised this as suggested.

Point 9: Line 223-224. Describe the stabilizing process as compared to an standard pelletization

Response 9: Thanks for your suggestion. We have revised this as suggested.

Point 10: Line 225-228. This comparison among the response of different vitamins to the pelletization process and the stabilizing process merits a more detailed discussion and published references on the sensitivity of vitamins to heat and pressure treatments.

In general: please use one writing, for instance ‘past’ or ‘present’ Donot mix these in the article as a whole

Response 10: Thanks for your suggestion. We have revised this as suggested.

Point 11: Tables: provide only 1 decimal for the average means, and 2 decimals for the SEM values (*alway one decimal more for the variability values than mean values)

Response 11: Thanks for your suggestion. We have revised this as suggested.

Round 2

Reviewer 1 Report

The authors effectively responded to all the concerns raised and I do not have any reservation as to why the manuscript should not be published.

Reviewer 2 Report

-